# A Personalized Approach to Treat Advanced Stage Severely Contracted Joints in Dupuytren’s Disease with a Unique Skeletal Distraction Device—Utilizing Modern Imaging Tools to Enhance Safety for the Patient

**DOI:** 10.3390/jpm12030378

**Published:** 2022-03-01

**Authors:** Wibke Müller-Seubert, Aijia Cai, Andreas Arkudas, Ingo Ludolph, Niklas Fritz, Raymund E. Horch

**Affiliations:** Department of Plastic and Hand Surgery, Friedrich-Alexander-University Erlangen-Nürnberg (FAU), Krankenhausstr. 12, 91054 Erlangen, Germany; aijia.cai@uk-erlangen.de (A.C.); andreas.arkudas@uk-erlangen.de (A.A.); ingo.ludolph@uk-erlangen.de (I.L.); niklas.fritz@uk-erlangen.de (N.F.); raymund.horch@uk-erlangen.de (R.E.H.)

**Keywords:** joint dislocation, Dupuytren contracture, distracted driving

## Abstract

Background: While surgical therapy for Dupuytren’s disease is a well-established standard procedure, severe joint flexion deformities in advanced Dupuytren’s disease remain challenging to treat. Skeletal distraction has proven to be an additional treatment option. Methods: We analyzed the surgical treatment algorithm, including the application of a skeletal distraction device, in patients with a flexion deformity due to Dupuytren’s disease, Iselin stage III or IV, who were operated on from 2003 to 2020 in our department. Results: From a total of 724 patients, we included the outcome of 55 patients’ fingers in this study, who had undergone additional skeletal joint distraction with our Erlangen device. Additional fasciotomy or fasciectomy, in a one- or two-staged procedure, was performed in all patients, according to the individual findings and necessities. The range of motion of the PIP joint improved from 12° to 53°. A number of complications, in all steps of the treatment, were noted in a total of 36.4% of patients, including the development of fractures (16.4%), followed by vessel injury, pin infections, and complex regional pain syndrome (5%). Conclusions: Additional skeletal distraction improves the range of motion of severely contracted joints in Dupuytren’s disease. Nevertheless, careful patient selection is necessary, due to the moderate rate of complications.

## 1. Introduction

While the therapeutic measures to surgically treat stage I and stage II Dupuytren’s disease (DD) are straightforward, and rely on neuro- and arterio-lysis with partial fasciectomy as a standard procedure, treating severe finger joint flexion deformities due to advanced Dupuytren’s disease still remains challenging [1]. There is no doubt that improving the extension of the contracted fingers is very important to restore hand function [2], and that postoperative physiotherapy is an essential cornerstone of successful treatment [3]. However, there has been an ongoing scientific debate as to whether arthrolysis should be attempted at all in severe and far advanced DD finger joint contractures, since the outcome is hampered by either relapse or skin necrosis and wound healing problems that may well end in partial or total finger loss [4,5,6,7,8]. The short-term results indicate temporary relief with enzymatic (collagenase) drug injections, but do not guarantee long-term success [9]. It is well known that the risk of a remaining or rapidly relapsing flexion deformity of the proximal interphalangeal (PIP) joint after an operation, due to shrinking, shortening, and/or adhesion of the periarticular structures, remains high [8], and increases with the degree of flexion contracture present before surgery [8]. It has been described that the complication rate increases with the severity of the disease, particularly if the proximal interphalangeal joint contracture was 60 degrees or more [7]. In many instances, joint fusion of the proximal and/or distal interphalangeal joint has been described as an appropriate remedy to achieve improved finger and, hence, overall hand function [7]. This is especially true for the little finger, where shortening through a dorsal approach seems to provide the following satisfactory outcomes: an acceptable looking functional finger that has sensation and no significant morbidity [10,11]. Attempts to reduce fourth degree finger joint flexion contractures in a single-step approach may end up in finger loss and amputation [4,12,13]. In order to minimize the risk and to improve the postoperative results when treating severe joint contractures, previous distraction of the affected joints has been reported using various devices. Among these, pneumatic pre-expansion [14,15] or skeletal distraction devices have been proposed [16,17,18,19,20,21,22]. The skeletal distraction device was initially developed by Messina [22]—the so-called “TEC” device—which can be considered a milestone in treating this entity. Utilizing an external fixator with adjustable distraction attachments, continuous elongation of skin and soft tissues, as well as ligaments and Dupuytren’s chords, can be achieved. According to Messina, this technique is meant to be a preparatory step for excision of the pathologic palmar fascia for severe Dupuytren’s contracture. The device consists of physiologic, painless, and atraumatic elongation that is obtained by means of a device fixed on the fourth and fifth metacarpal bones by two self-drilling pins [22] (Figure 1).

Albeit this device is a proper means to tackle severe finger joint contractures, its use turned out to be irritating and cumbersome over longer time periods, due to its considerably bulky size and skin irritation at the site of the distal transphalangeal wires. In addition, Messina’s idea that pretreatment with a TEC device would also avoid any correction of digital or palmar skin loss by plastic surgery was not quite achieved in all cases.

As a single treatment in those patients, after removal of the Messina device, rapid recurrence followed, but its use as a prelude to surgery improved the final results, and the tissue dissection during secondary fasciectomy was described to be easier [23]. It received further advancements or modifications, for instance as “Pipster” by Hodgkinson [20], and was further miniaturized by our group with a turning worm-screw modification, termed an Erlangen distraction device (Figure 2) [19].

By slow continuous distraction of the retracted finger joints, any sudden stretching and tearing of the shortened neurovascular bundles is avoided. Abrupt and harsh stretching of the finger vessels is one of the main causes of devascularization and ischemic disorders, with possible subsequent necrosis, sensation loss, and, ultimately, finger loss [13,22,24]. Furthermore, staged approaches with distraction of severely contracted joints, followed by fasciectomy, have been shown to minimize complication rates [22,25]. In contrast, minimally invasive methods, such as collagenase injections and percutaneous needle transection, are not suitable to treat advanced joint contractures [26,27,28]. The hypothesis of this study was to evaluate the benefit and the complications of an external skeletal distraction device in the treatment of severe joint contractures in DD. A prospective study, or a comparison of the results with a control group, is not feasible in this context, since every patient with such extremely progressed contractures received an individualized specific treatment for his or her severe end-stage contracted finger joint, due to DD.

## 2. Methods

We analyzed all surgically treated patients (*n* = 724 fingers) and included those with advanced DD Iselin stage III–IV who received a skeletal distraction device to treat flexion joint deformities between the years 2003 and 2020 in our Department of Plastic and Hand Surgery, to identify the most appropriate personalized approach for this problematic entity. The study was approved by the institutional ethics committee (ethics committee vote number 177_20 Bc), and STROBE guidelines were followed. The Strengthening the Reporting of Observational Studies in Epidemiology (STROBE) Initiative developed recommendations on what should be included in an accurate and complete report of an observational study.

We applied our skeletal distraction device to the contracted fingers with self-drilling pins under X-ray control, as previously described elsewhere [19,29], and dissected the main Dupuytren chord in the palm during the first step to allow metacarpophalangeal joint mobilization. This was accompanied by Z-plasty to achieve skin and soft tissue elongation in the palm. While still under local anesthesia, the distraction device was probatorily stretched until blanching of the skin was noted. To determine the amount of initial stretching, we then adjusted the degree of extension using hyperspectral imaging and thermography. By applying these contact-free measuring tools, the oxygen saturation and hemoglobin view, as well as the microperfusion pattern, can be directly visualized, and, hence, the stretching is reduced to a degree where the finger perfusion is within the normal and safe range. These relatively new modalities have been described to evaluate perfusion patterns and areal oxygen saturation [30,31]. By doing so, we can easily determine the maximum degree of pre-extension within the margins of safe microcirculation, and find out how far we can safely extend the distractor at that initial moment, without jeopardizing microcirculation (Figure 3). When we noticed blanching and deterioration of finger microcirculation, we released the distractor until microcirculation returned to normal values. This can easily be visualized and measured, as is shown in Figure 3. In case of any doubt about microcirculation, one could also apply indocyanine green near-infrared angiography to determine aterial influx in the finger [32].

This makes the whole procedure safer, in terms of maintaining sufficient microcirculation and not harming the blood supply to the extended finger. The patient is advised to distract the device as slowly as possible so that she or he should not perceive pain. The programmed distraction is then started within 3–5 days postoperatively, and is commenced according to the individual tension of the contracted joints. The end point of distraction is achieved when the device is fully stretched to zero degrees, or when maximum distraction is obtained.

The data were recorded retrospectively using in- and out-patient medical records. Main fields of interest were the period of distraction as well as any intra- and post-operative complications, such as fractures, injury of vessels, development of complex regional pain syndrome (CRPS type I), as well as Boutonniere deformity and infection. Furthermore, basic data, such as gender, age and follow-up period, were recorded. A descriptive statistical data analysis was performed using Prism 8 (GraphPad Software, San Diego, CA, USA).

## 3. Results

Between January 2003 and May 2020, a total of 724 fingers with DD were surgically treated in our Department of Plastic and Hand Surgery (Figure 4). During this period, 55 fingers of 53 patients with severe DD Iselin grade III–IV (Figure 5 and Figure 6) were operated upon.

In total, 41 men and 12 women were included in this study. The average age of the patients was 64 years (range 35–89 years). The average follow-up period was 19 weeks (range 0–380 weeks).

Thirty-one fingers were treated with a staged approach of fasciotomy and skeletal distraction, followed by partial fasciectomy with zigzag-shaped skin incisions [33]. Eleven fingers were satisfactorily treated with palmar fasciotomy and skeletal distraction. Thirteen fingers received simultaneous skeletal distraction combined with partial fasciectomy (Figure 7, Figure 8 and Figure 9).

The distraction device was removed after an average period of 28 days (range 12–66 days), when full or near-full extension was achieved. A distraction period of 66 days involuntarily occurred in one patient, as a result of the COVID-19 pandemic and a verdict to halt elective operations at that time, so the second step operation was postponed.

In all fingers, the range of motion of the PIP joint improved from an average of 12° (extension/flexion 0–74–86°) to 53° (extension/flexion 0–20–73°). To further describe the results in combination with the treatment options, we split the data into subgroups. Hence, in the two-staged group, the intraindividual range of motion of the PIP joint improved from 0° (extension/flexion 0–76–76°) to 25° (extension/flexion 0–23–48°). Further subgroup analysis did not yield sufficiently significant data, due to variable and incomplete data sets.

In the total number of patients, the overall complication rate, including minor and major side-effects (Table 1), was found to be 36.4% (*n* = 20/55 fingers).

Injury to vessels was noted in three patients (5.5%), during the secondary fasciectomy. In three cases (5.5%), early removal of the distraction device was necessary because of infection or traumatic distortion, with one patient developing an infection 6 weeks after removal of the distraction device, which required successful surgical revision. A total of six patients developed a fracture of the middle phalanx, two patients (3.6%) of the end phalanx, and one patient (1.8%) of the proximal phalanx during distraction. Those patients who developed a fracture had a slightly higher age of 67 years compared to the overall collective. While some fractures were treated with additional K-wire osteosynthesis, the others were treated conservatively, and all healed well, without further clinical problems. One patient with a recurrent 80° Dupuytren’s flexion contracture of the PIP joint was treated with arthrodesis 9 months after second stage fasciectomy, and was very content with the final result. Similar to other hand surgery patients, three patients developed a complex regional pain syndrome (CRPS I). Postoperative hematoma, after removal of the distraction device, was observed in one patient, who then required operative treatment. The development of a Boutonniere deformity of the treated finger was observed after one month in one patient. A patient with a rupture of the deep flexor tendon of the distracted finger in his medical record was also observed. He had previously been treated in another hospital four times because of relapsing DD in his little finger, followed by an infection and wound healing disorder, before presenting at our hospital. This patient showed a flexion deformity of 90° of the PIP joint of his little finger, with extensive palmar scarring and skin shortening. After a short period of skeletal distraction, fasciectomy, together with a cross finger flap from the ring finger, were performed. In addition, 4.5 months after the end of treatment, the patient regained sufficient finger function.

## 4. Discussion

Although alternative measures to circumvent surgery in Dupuytren’s disease have been repeatedly investigated, operative procedures are the cornerstone of treatment when functional loss occurs. It is necessary to personalize the therapeutic approach according to the individual stage of disease and functional impairment, as well as the patient’s general condition and demands. While most patients present with stage I or stage II and can routinely be operated upon with partial fasciectomy and arthrolysis, the treatment of severe flexion finger joint contractures in DD still remains a challenge. It is necessary to offer the full range of options to best address the individual needs, in order to balance the up and down sides of various surgical options. New techniques, such as the staged distraction of finger joint contractures, can help to expand the array of therapeutic possibilities for the patient.

Despite recent advances, the process of proliferation of (myo-) fibroblasts is not yet understood [34,35,36,37,38,39] and methods of tissue engineering and regenerative medicine to replace resected tissue and skin are promising, but are still not clinically available [37]. While MCP contracture may be well addressed with segmental palmar aponeurectomy with Z-plasty [40], PIP and DIP joint contractures are more difficult to treat [41]. In an anatomical study, severe secondary damage to lateral and dorsal structures was observed, and these changes may explain the poor results of corrective surgery to this joint in DD [42]. Opening long-standing severe contractions may lead to an impairment of microperfusion, and, consequently, to necrosis. Newer methods to intraoperatively distinguish the true amount of cutaneous microcirculation, similar to other areas of surgery, might help to reduce such sequelae [43,44]. The application of topical negative pressure to the closed incision could potentially optimize perfusion [45].

It has been found that the degree of PIP joint contracture is related to the outcome of surgical treatment of DD. Effective relief of joint extension inhibition is mostly achieved when the contracture degree is between 15° and 30°. Furthermore, the complication rate increases with the severity of the disease, especially when the PIP joint contracture is 60° or more [46]. To distract the contracted joint step by step, Messina used an external fixation device to stretch the contracted skin, as well as the contracted joint structures [22]. Multiple modifications of skeletal distraction devices have already been described, including the miniaturized Erlangen traction device [16,17,18,19,20,21]. The main difference is that the Messina device is too bulky, creates skin lacerations at the wiring sites, and does not allow normal clothing to be worn. In addition, it does not effectively distract the joints to zero degree extension. Maybe this is the reason why this device has not gained widespread application. The device used in this study is as small as possible. It attacks the joints more closely to the constricted ligaments, and acts more powerfully and directly at the joint. The self-drilling pins are far more stable than Messina’s wires. Thus, the distraction force is immediately transmitted to the relevant bony structures. It is far more convenient and user friendly, and allows normal clothing to be worn.

Slow continuous distraction of the affected joints has become a standard procedure in our department to avoid injury to the nerves and vessels of severely contracted joints of more than 75° in DD. Our patients gained an average improvement of motion of the treated PIP joint of 53°. White et al. improved the PIP joint flexion deformity from 75° to 37° by a two-staged procedure with a mini external fixator, followed by fasciectomy after 6 weeks of distraction [21]. Treatment with single stage fasciectomy and the “S Quattro” distraction device improved the arc of motion from 10° preoperative to 34° postoperative [47]. In comparison, the two-staged approaches of other studies, as well as from our study, seem to lead to a slightly better postoperative range of motion.

Complications in any surgical treatment of DD have been described in 18.2% of patients in a prospective study by Bulstrode et al. [46]. The main complication in our study was the development of fractures of the distracted fingers. Interestingly, none of the other studies reported fractures during the distracting period [17,21,22,47,48]. In our study, application of the skeletal distraction device was performed carefully by using X-ray. One patient fell on his distraction device, which might have been the cause of the fracture. Two patients developed a fracture of the distal phalanx, even though the DIP joint was not distracted. One of those patients presented a deep flexor tendon injury after a previous infection elsewhere, so the fracture might have been a complication related to the previous operative therapy. The second patient received transfixation of the DIP joint after fasciectomy because of contracture in the DIP joint as well. It is assumed that the tension on the ankylotic DIP joint was too strong, and this might have resulted in a fracture. Skeletal distraction in the opposite direction to the tension of the contracted joint might lead to an injury of the weakest structure, in this case, the bone. Furthermore, the average age of patients who developed fractures during the distraction period was slightly higher compared to the age of the total collective, so osteopenia might be another reason for the development of fractures. One should be aware that treatment of an additional fracture requires further splinting, in addition to potential operative treatment. Further splinting of former extremely contracted and often stiff joints increases the risk of remaining stiffness, or, at least, reduces the range of motion.

Our rate of pin infections of 5.5% was low compared to the 16.7% [47] or 21% [21] reported by other studies. Our patients received preventive antibiotic treatment, while a postoperative medication regime was not described in other studies. Messina et al. did not report any infections, or vascular or sensation deficits, in their patients [22]. Our careful distraction over a longer period of 28 days, compared to 14 days [22,47] or 19 days [17], resulted in no vascular compromise during the distraction period, in contrast to 5.6% in the study by Beard et al. [47]. Our rate of vessel injury during fasciectomy was similar to the rate of vessel injury of 4.1% after fasciectomy and 5.6% after dermofasciectomy [49]. The rate of chronic regional pain syndrome (CRPS) after skeletal distraction was no higher than the rate after single fasciectomy (10.2%) or dermofasciectomy (4.1%) without any distraction device [13,49]. Due to the large number of patients and the long period of time, it is unfortunately not possible to make a precise statement regarding the formation of hypertrophic scars.

Similarly to our study, White et al. reported additional arthrodesis in 2.6% of fingers after two-staged treatment [21]. As described by others, to avoid further deterioration of the neurovascular bundles following previously unsuccessful palmar operative treatments in severe finger joint contractures, we performed joint fusion via a dorsal approach, with resection of the joint, and, thus, shortened the length of the finger. By doing so, the already shortened or compromised neurovascular bundles and the soft tissue within the scarred palmar envelope were touched [48]. When arthrosis of the PIP joint is observed, due to a long-lasting flexion deformity, finger joint arthroplasty is not an option because of shrinkage of the ligaments [50,51]. Intraoperative rapid distraction may be helpful, but will probably not last for long [52]. Adjusting the degree of extension using hyperspectral imaging and thermography is another tool that allows the microcirculation to be monitored when a contracted finger is distracted. Thus, the initial tension at the beginning of distraction can be individually adjusted, and distortion of the microcirculation can be circumvented. The fact that we observed a moderate risk of developing a fracture during the distraction period was associated with the age of the elderly patients with long-standing finger joint ankylosis. This might be due to osteoporosis and the often very long standing shrinkage of the ligaments. Therefore, if osteoporosis is visible, this may pose a contraindication for our device. In a patient who suffered a finger fracture during distraction, the fracture healed uneventfully and in a sufficient position, so the patient was very content with the result. We conclude that, under such circumstances, finger arthrodesis should be discussed as a valuable alternative. Nevertheless, given the ease of application and the painless slow distraction phase, in principle, all stage III and stage IV finger joint contractures are eligible for the Erlangen device.

The possible limitations of this study include the necessarily retrospective study design for a rare group of patients with extremely advanced stages of DD, and, hence, the lack of a control group. Since each patient was treated individually, according to their DD stage and localization, comparison of our results with a control group was not possible. The follow-up period was divergent, and since the patients came from abroad for this specialized treatment, some could not be followed up. This was mostly due to the distances that patients with far advanced finger joint contractures had to travel, who were mostly secondarily referred to our specialty service.

## 5. Conclusions

Various stages of DD need various surgical approaches, which must be indicated according to the extent of the individual patient’s disease. In far advanced and relapsing stages of DD, external skeletal distraction of severe joint flexion deformities is a feasible and valuable tool to (pre-)extend contracted finger joints carefully and slowly. It adds another option for treating these extremely functionally restricted hands. Since the treatment of far advanced contractures can be highly challenging, careful selection and education of the patients, as well as repeated and timely follow-up examinations, are necessary. Distraction of ankylotic joints in flexion deformity in elderly patients should also be indicated carefully, due to osteopenia, but it holds promise for functional benefits in elderly patients.

## Figures and Tables

**Figure 1 jpm-12-00378-f001:**
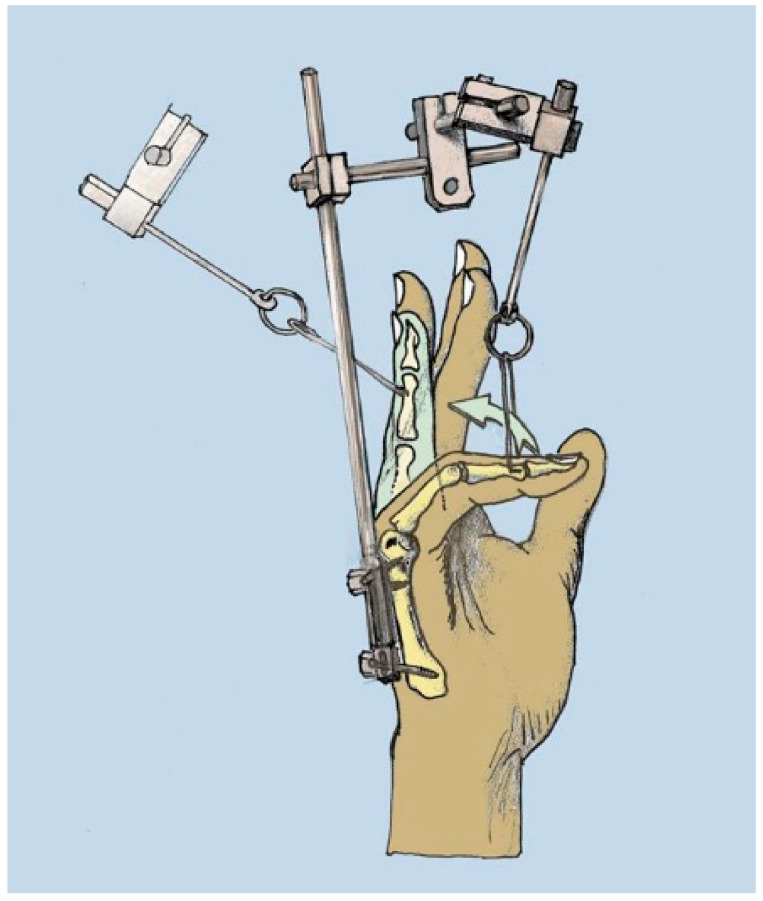
Schematic depiction of Messina’s skeletal distraction device, modified from Brenner P, Ray Ghazi M. (ed.) Morbus Dupuytren: Ein chirurgisches Therapiekonzept, Springer Verlag Berlin-Heidelberg-New York, 2003, ISBN 978-3-7091-6723-6 (copyright R E Horch).

**Figure 2 jpm-12-00378-f002:**
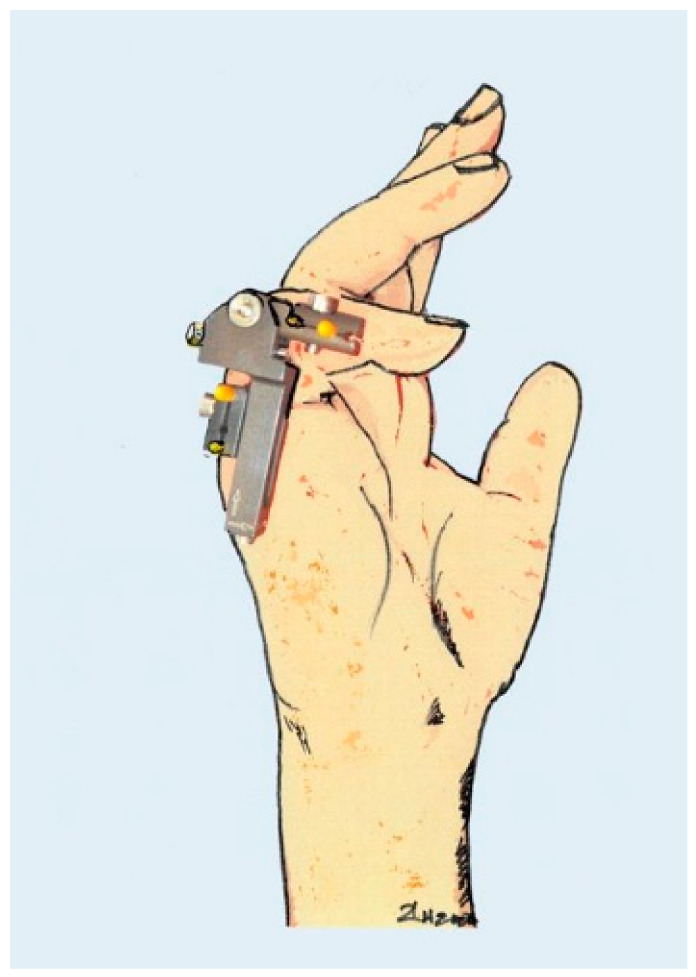
Erlangen miniaturized distraction device with turning worm-screw mechanism (copyright R E Horch 2020).

**Figure 3 jpm-12-00378-f003:**
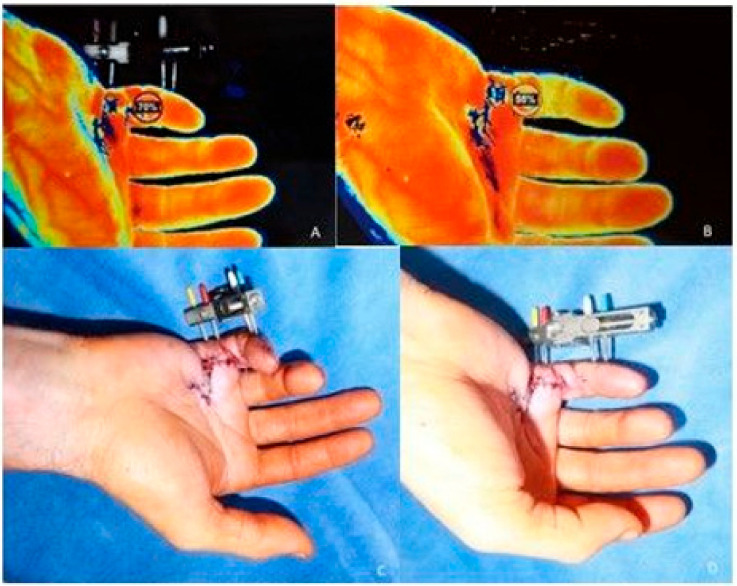
Hyperspectral imaging showing malperfusion (yellow parts) after extension subfigure (**B**) in comparison to the non-extended finger subfigure (**A**). Red color represents well-perfused skin, as measured with the no-contact hyperspectral imaging device, consistently distributed over the visible palm of the hand. A change in optimal oxygen saturation (red = 100% oxygen saturation) is shown as yellow, representing decreasing amounts of oxygen saturation. Under distraction, more yellow can be observed in the right depiction (**B**). Corresponding clinical images of the non-extended (**C**) and extended finger (**D**).

**Figure 4 jpm-12-00378-f004:**
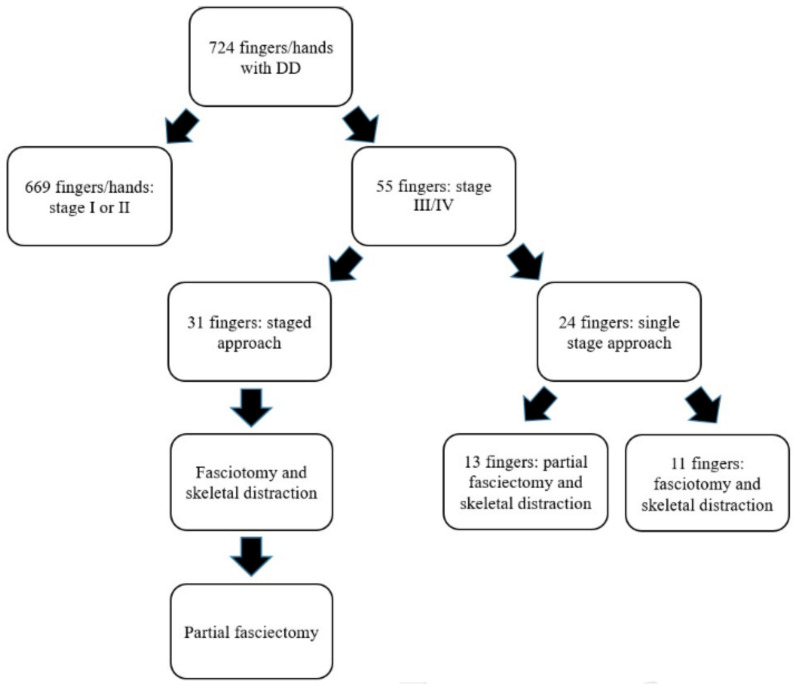
Individualized treatment regimen of DD.

**Figure 5 jpm-12-00378-f005:**
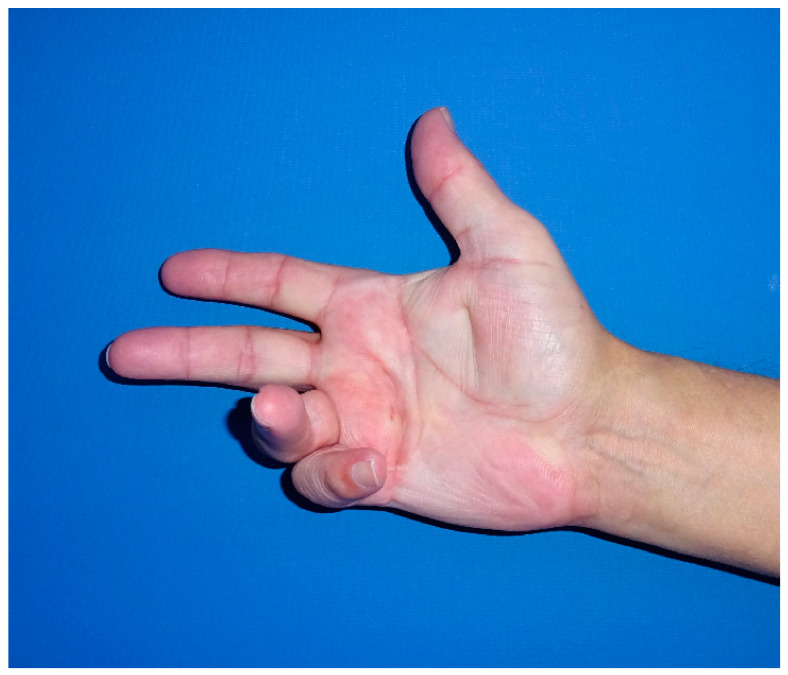
Patient with stage IV DD of the little finger.

**Figure 6 jpm-12-00378-f006:**
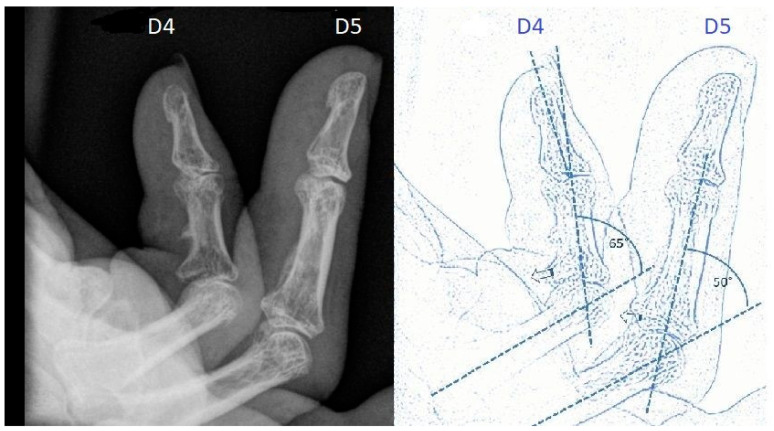
Preoperative X-ray of stage IV flexion deformity and schematic drawing.

**Figure 7 jpm-12-00378-f007:**
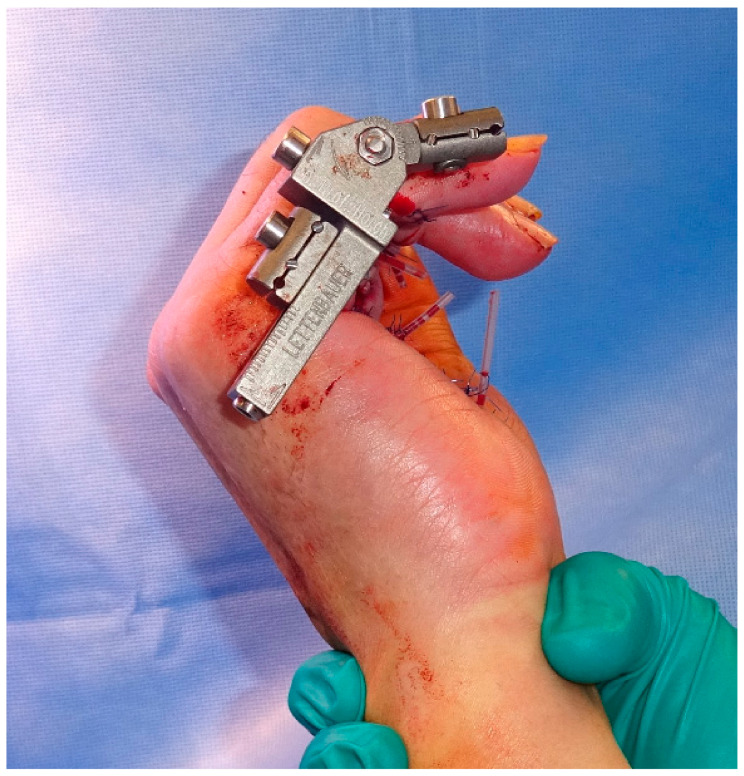
Intraoperative application of the skeletal distraction device in combination with fasciectomy.

**Figure 8 jpm-12-00378-f008:**
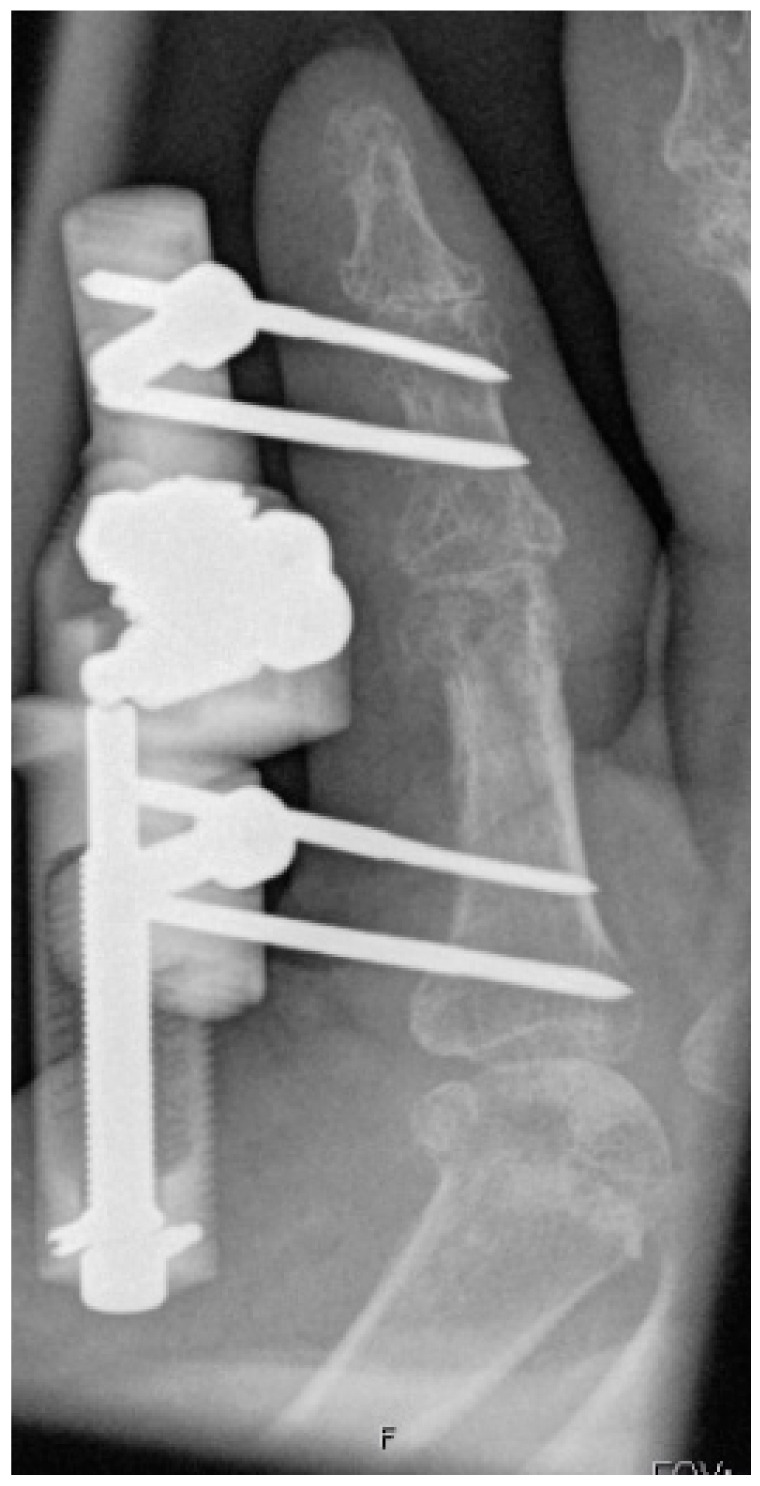
Completely extended PIP joint of the same patient shown in Figure 5, Figure 6 and Figure 7 at 6 weeks of distraction showing complete release of the PIP joint.

**Figure 9 jpm-12-00378-f009:**
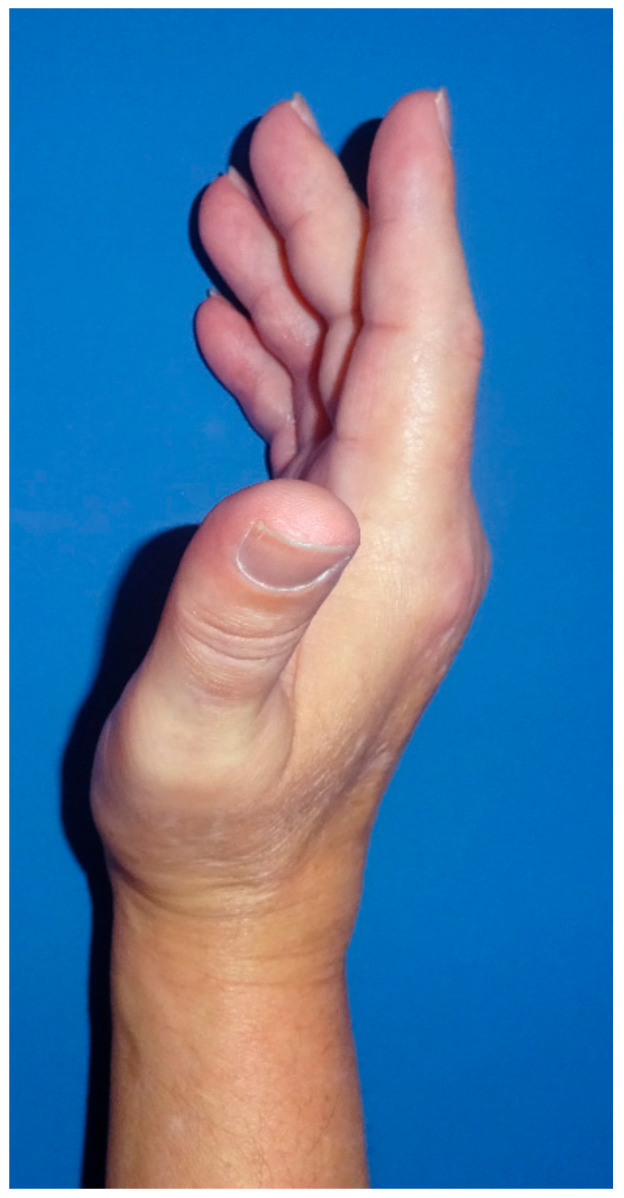
Presented patient 8 weeks after fasciectomy.

**Table 1 jpm-12-00378-t001:** Complications.

Complications	Percentage [%]
Vessel injury	5.5
Pin infection	5.5
Infection	1.8
Fracture	16.4
Arthrodesis	1.8
CRPS I	5.5
Hematoma	1.8
Boutonniere deformity	1.8
Flexor tendon injury	1.8
These complications occured in 20 fingers	

## Data Availability

Data sharing is not applicable to this article.

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
