# Peer review of "A Personalized Approach to Treat Advanced Stage Severely Contracted Joints in Dupuytren’s Disease with a Unique Skeletal Distraction Device—Utilizing Modern Imaging Tools to Enhance Safety for the Patient"

_jpm, 2022, doi:10.3390/jpm12030378_

Round 1

Reviewer 1 Report

  1. This sentence is not clear, please modify:

“A number of complications of all steps of treatment were noted in a total of 36.4% including the development of fractures in 16.4 followed by vessel injury, pin infections and complex regional pain syndrome (5 %).”

  1. This is not clear, please modify:

“In the past decades, severe flexion deformities of finger joints resulted not seldom in amputations of the affected fingers [3]”

  1. In the sentence, “It should be noted that the risk of a remaining or rapidly relapsing flexion deformity of the PIP joint after an operation due to shrinking, shortening, and/or adhesion of the periarticular structures increases with the degree of flexion contracture present before surgery is extremely high [3].” – the “is extremely high” is not required.
  2. Please avoid using “not seldom” because the sentences are not clear. Please change them in the text.
  3. Please be more technical in explaining the drawbacks of the TEC device.
  4. Please correct spelling of “ferasible”.
  5. In the Methods section, please correct the language, “We analyzed all surgically patients”
  6. The authors should specify what 177_20 Bc and STROBE guidelines stand for
  7. The authors should specify how they apply their skeletal distraction device in detail and how the degree of extension is adjusted using their device. Details regarding how the hyperspectral imaging and thermography is used should be provided.
  8. “Accordingly distraction is started within 3 - 5 day according to the individual tension of the contracted joints” – the authors should provide more details regarding this criterion.
  9. The authors mention that the follow-up period for their study was 19 weeks (range 0-380 weeks). Is 19 weeks the average follow-up period?
  10. Figure 8 should be labeled and compared with the original untreated joint.
  11. The authors mention that the in all fingers the range of motion of the PIP joint improved from an average of 12° to 53°. Is this in the one-stage or two-stage treatment?
  12. What were the differences in results between the one-staged and two-staged treatments?
  13. The authors should specify how they calculated the individual complication percentages.
  14. How is the overall complication rate 36.4%? It is unclear what this group refers to and how n = 20 for this data set. These details should be clarified.
  15. The authors should correct this, “a history rupture of the deep flexor tendon”.
  16. In the last paragraph of the Results section, it is not clear whether these results refer to the one-staged or two-staged treatments and the correlation with percentages between the text and Table 1 is not clear. The authors should revise this paragraph to incorporate these changes.
  17. The authors should provide the full form of abbreviations they are using when they are mentioned in the text.
  18. It is not clear what the difference is between various modifications of different skeletal distraction devices and the device used by the authors and what advantages, if any, is offered by the device used by the authors.
  19. The authors mention that their patients gained an average improvement of motion of the treated PIP joint of 50°. Is this from this study? It is mentioned as 53° previously. Please be consistent.
  20. This sentence is not clear, please modify, “The shortening arthrodesis protects the shortened neurovascular bundle and soft tissue release of the previously operated and scarred palmar envelope is not necessary due to the bony resection”.

Author Response

Reply to Reviewer comments:

Reviewer 1:

Point 1: This sentence is not clear, please modify:

“A number of complications of all steps of treatment were noted in a total of 36.4% including the development of fractures in 16.4 followed by vessel injury, pin infections and complex regional pain syndrome (5 %).”.

Answer 1: Thank you for this hint, we revised this passage

Point 2: This is not clear, please modify:

“In the past decades, severe flexion deformities of finger joints resulted not seldom in amputations of the affected fingers [3]”

Answer 2: Thanks, we revised this passage

Point 3: In the sentence, “It should be noted that the risk of a remaining or rapidly relapsing flexion deformity of the PIP joint after an operation due to shrinking, shortening, and/or adhesion of the periarticular structures increases with the degree of flexion contracture present before surgery is extremely high [3].” – the “is extremely high” is not required.

Answer 3: This is correct, we revised this passage

Point 4: Please avoid using “not seldom” because the sentences are not clear. Please change them in the text.

Answer 4: thank you! we revised this sentence

Point 5: Please be more technical in explaining the drawbacks of the TEC device.

Answer 5: This is correct, we revised this passage anddescribed the Messina device appropriately

Point 6: Please correct spelling of “ferasible”.

Answer 6: Thank you, we corrected this misspelling

Point 7: In the Methods section, please correct the language, “We analyzed all surgically patients”

Answer 7: thank you! we revised this sentence

Point 8: The authors should specify what 177_20 Bc and STROBE guidelines stand for

Answer 8: thank you! we explained these shortened acronyms in the revised manuscript  ( the first one is the registration number of the ethics committee´s vote, the second one is explained in the revised text)

Point 9: The authors should specify how they apply their skeletal distraction device in detail and how the degree of extension is adjusted using their device.

Answer 9: We have previously described the application of our device and
Details regarding how the hyperspectral imaging and thermography is used.
Thank you, we have added these details in our revision

Point 10: “Accordingly distraction is started within 3 - 5 day according to the individual tension of the contracted joints” – the authors should provide more details regarding this criterion.

Answer 10: thank you! we revised this sentence and we have added these details in our revision

Point 11: The authors mention that the follow-up period for their study was 19 weeks (range 0-380 weeks). Is 19 weeks the average follow-up period?

Answer 11: Thank you for this remark, We have reported the „average“

Point 12: Figure 8 should be labeled and compared with the original untreated joint.

Answer 12: Figure 8 shows the same patient as in figure 5, 6 and 7 and we corrected the legend accordingly

Point 13: The authors mention that the in all fingers the range of motion of the PIP joint improved from an average of 12° to 53°. Is this in the one-stage or two-stage treatment?

Answer 13: First we described the overall results in all patients. Then we specified the data accordingly between the subgroups. [……… in the two-staged group, the intra individual range of motion of the PIP joint improved from 0° (extension/flexion 0-76-76°) to 25° (extension/flexion 0-23-48°). Further subgroup analysis did not yield sufficiently significant data due to variable and incomplete data sets….]

Point 14: What were the differences in results between the one-staged and two-staged treatments?

Answer 14: Unfortunately, subgroup analysis did not yield sufficiently significant data due to variable and incomplete data sets.

Point 15: The authors should specify how they calculated the individual complication percentages.

Answer 15: We summarized any possible complication and reported all the complications divided by the number of treated patients, (results in % of total numbers).

Point 16: How is the overall complication rate 36.4%? It is unclear what this group refers to and how n = 20 for this data set. These details should be clarified.

Answer 16: Thank you for this hint: we accordingly explained these data in the revision:
………In the total number of our patients the overall complications rate, including minor and major side effects, (Table 1) was found to be 36.4% (n=20 fingers/55 fingers).

Point 17: The authors should correct this, “a history rupture of the deep flexor tendon”.

Answer 17: Thank you, we corrected this one.

Point 18: In the last paragraph of the Results section, it is not clear whether these results refer to the one-staged or two-staged treatments and the correlation with percentages between the text and Table 1 is not clear. The authors should revise this paragraph to incorporate these changes.

Answer 18: As we corrected this in the revision, data belong to both groups, and further subgroup distinction was not applicable due to a lack of statistical power.

Point 19: The authors should provide the full form of abbreviations they are using when they are mentioned in the text.

Answer 19: Thank you we have revisited the manuscript and explained any abbreviations

Point 20: It is not clear what the difference is between various modifications of different skeletal distraction devices and the device used by the authors and what advantages, if any, is offered by the device used by the authors.

Answer 20: Thanks we added this explanation to the manuscript.

Point 21: The authors mention that their patients gained an average improvement of motion of the treated PIP joint of 50°. Is this from this study? It is mentioned as 53° previously. Please be consistent.

Answer 21: Thank you you are right, we have corrected this data.

Point 22: This sentence is not clear, please modify, “The shortening arthrodesis protects the shortened neurovascular bundle and soft tissue release of the previously operated and scarred palmar envelope is not necessary due to the bony resection”.

Answer 22: Thank you, we have corrected this passage

Reviewer 2 Report

  1. Figures could be displayed more neatly. As shown, Figure 1 and Figure 2 can be combined with description, Figure 6-9 can be combined to display.
  2. I'm still very curious about what type of patients would be suitable for applying Erlangen external distraction device? Would all the stage III or IV patients, based on the authors' results?
  3. Statistical analysis needs to be described in more detail.
  4. The content of Figure 3 is not easy to understand, please indicate.
  5. How about other Complications after treatment, likely Hypertrophic Scars or Numbness in the fingers?

Author Response

Point 1: Figures could be displayed more neatly. As shown, Figure 1 and Figure 2 can be combined with description, Figure 6-9 can be combined to display.

Answer 1: We have filled in the figures in a running fashion and trust that the journal will  as always – arrange the figures accordingly.

Point 2: I'm still very curious about what type of patients would be suitable for applying Erlangen external distraction device? Would all the stage III or IV patients, based on the authors' results?

Answer 2: Thank you very much. We have added a specific passage to better explain the indication. Given the ease of application and the painless slow distraction phase all stage III and stage IV finger joint contractures are eligible for the Erlangen device. However, patient compliance is essential, so that a lack of compliance would be a contraindication. Far advanced osteoporosis might also be a contraindication, s fractures might occur. In a patient who underwent a finger fracture this healed uneventful and in a sufficient position so that the patient was very content with the result.

Point 3: Statistical analysis needs to be described in more detail.

Answer 3: Thank you for this hint! As we explained in the manuscript after discussing the raw data with our statisticians we felt that due to the various subgroups a true statistical analysis would not prove to underline the findings. Given the fact-  as so often in surgery -  that this was a retrospective data analysis only [over a period of almost two decades] that pre- and postoperative data were inconsistently reported and documented we could only extract such data that were truly comparably documented and therefore concentrated on the description of our findings rather than performing statistical analyses that would not really significantly add more value to our findings.
Even in a specialized center for Dupuytren´s disease such as our unit (we have published a series of more than 3000 DD patients previously), stage III and stage IV DD contractures are fortunately scarce when compared to “ regular“ patients. Despite the obvious obstacles we plan a prospective study to more exactly define the exact data of distraction and joint mobilisation in the  future. However this might well take another 20 years of research, until we can come up with such data.

Nevertheless, we believe that to the best of our knowledge this is by far the largest cohort of only stage III and stage IV DD patients with severe finger joint contractures with this device up to now. (Piza Katzer reported on 10 patients, Brandes and Messina on 4 patients, K Beyermann  , C Jacobs, K-J Prommersberger, U Lanz on 9 patients), so that we believe that communicating these findings should be helpful for decision making and for an individualized approach to this special topic.

Point  4: The content of Figure 3 is not easy to understand, please indicate.

Answer 4: Thank you for the remark; we indicated this in the revision and replaced the figure and the caption.

Point 5: How about other Complications after treatment, likely Hypertrophic Scars or Numbness in the fingers?

Answer 5: no hypertrophic scars were seen and no nerve injuries were documented.

Round 2

Reviewer 2 Report

  1. In your previous report ‘Med Sci Monit. 2021 Apr 22;27:e929814‘. Hypertrophic scars appeared most frequently in 9 of 13 hands (69%) after treating with Erlangen external distraction device. How comes no hypertrophic scars were seen nor documented in this study?
  2. Table 1 should be displayed as a three-line form. 

Author Response

Point 1: In your previous report ‘Med Sci Monit. 2021 Apr 22;27:e929814‘. Hypertrophic scars appeared most frequently in 9 of 13 hands (69%) after treating with Erlangen external distraction device. How comes no hypertrophic scars were seen nor documented in this study?

Answer 1: Thank you for the comment. You are right, in the previous study the patients filled out the DASH score as well as dedicated follow-up examinations were performed.  This is the reason why hypertrophic scarring was noticed. In the current study, due to the long period of time and the large number of patients, it was unfortunately not possible to conduct such a dedicated follow-up examination. In contrast, the current study concentrates more on the general use and indications of the Erlangen distraction device in Dupuytren´s Disease.

We added the following sentence “Due to the large number of patients and the long period of time, it was unfortunately not possible to make a precise statement regarding the formation of hypertrophic scars.”

Point 2: Table 1 should be displayed as a three-line form.

Answer 2: Thank you for this hint. We changed the table.